# Smp24, a Scorpion-Venom Peptide, Exhibits Potent Antitumor Effects against Hepatoma HepG2 Cells via Multi-Mechanisms In Vivo and In Vitro

**DOI:** 10.3390/toxins14100717

**Published:** 2022-10-21

**Authors:** Tienthanh Nguyen, Ruiyin Guo, Jinwei Chai, Jiena Wu, Junfang Liu, Xin Chen, Mohamed A. Abdel-Rahman, Hu Xia, Xueqing Xu

**Affiliations:** 1Department of Pulmonary and Critical Care Medicine, Zhujiang Hospital, Southern Medical University, Guangzhou 510280, China; 2Guangdong Provincial Key Laboratory of New Drug Screening, School of Pharmaceutical Sciences, Southern Medical University, Guangzhou 510515, China; 3Zoology Department, Faculty of Science, Suez Canal University, Ismailia 41522, Egypt

**Keywords:** antitumor peptide, membrane disruption, mitochondrial dysfunction, apoptosis, cell cycle arrest, autophagy

## Abstract

Scorpion-venom-derived peptides have become a promising anticancer agent due to their cytotoxicity against tumor cells via multiple mechanisms. The suppressive effect of the cationic antimicrobial peptide Smp24, which is derived from the venom of *Scorpio*
*Maurus palmatus*, on the proliferation of the hepatoma cell line HepG2 has been reported earlier. However, its mode of action against HepG2 hepatoma cells remains unclear. In the current research, Smp24 was discovered to suppress the viability of HepG2 cells while having a minor effect on normal LO2 cells. Moreover, endocytosis and pore formation were demonstrated to be involved in the uptake of Smp24 into HepG2 cells, which subsequently interacted with the mitochondrial membrane and caused the decrease in its potential, cytoskeleton reorganization, ROS accumulation, mitochondrial dysfunction, and alteration of apoptosis- and autophagy-related signaling pathways. The protecting activity of Smp24 in the HepG2 xenograft mice model was also demonstrated. Therefore, our data suggest that the antitumor effect of Smp24 is closely related to the induction of cell apoptosis, cycle arrest, and autophagy via cell membrane disruption and mitochondrial dysfunction, suggesting a potential alternative in hepatocellular carcinoma treatment.

## 1. Introduction

Scorpions and their venom have a long history of serving people as traditional medicine against multiple conditions. Specially, scorpion-derived peptides are proved to become a promising remedy against severe diseases such as cancer and immune-related disease [1,2]. The active peptides from scorpions generally belong to disulfide-bridged peptides (DBPs) and non-disulfide-bridged peptides (NDBPs). As the important components of scorpion venom, DBPs are associated with neurotoxicity, while NDBPs are multiple functional cationic peptides possessing antibacterial, antiviral, anticancer, and immune-modulatory activities [3]. These peptides may serve as potential candidates for drug development with a broad range of applications [4]. Due to the need for novel anticancer drugs with low cytotoxicity and treatment resistance, antimicrobial peptides (AMPs) can represent a potential therapy by selectively targeting tumor cells through binding the negative-charged phosphatidylserine and syndecans on their surface, causing cell death by multiple mechanisms [5]. For instance, MSP-4, an AMP identified from the Nile tilapia (*Oreochromis niloticus*), has been described to decrease the viability of the human osteosarcoma cell line MG-63 through induction of cell cycle arrest and apoptosis by activating a Fas/FasL- and mitochondria-mediated pathway [6]. The cationic AMP reported from the hemocyanin of *Litopenaeus vannamei*, LvHemB1, inhibits the viability of various cancer cell lines while not affecting the normal liver cell lines [7]. Moreover, LvHemB1 exerts its antitumor effect by causing mitochondrial membrane potential loss, increasing reactive oxygen species (ROS) production and apoptotic proteins. We previously reported Smp24 (IWSFLIKAATKLLPSLFGGGKKDS), a cationic AMP identified from the venom of *Scorpio*
*Maurus palmatus* with a wide antimicrobial spectrum against various bacteria and fungi [8]. The cytotoxicity of Smp24 against two acute leukemia cell lines (KG1-a and CCRF-CEM), as well as four lung cancer cell lines (A549, H3122, PC-9, and H460) and nontumor cell lines (HRECs, CD34^+^, MCR-5, and HaCaT) has been described in further studies [9,10,11]. Smp24 also has cytotoxic effects against HepG2 hepatoma cells, as presented in ATP release assays; nevertheless, its mode of action against HepG2 hepatoma cells remains unclear. In this study, we explored the cytotoxicity of Smp24 and its impacts on the cell membrane, cell cycle distribution, apoptosis, and autophagy of HepG2 cells. Our results reveal that the antitumor activity of Smp24 is related to membrane disruption and mitochondrial dysfunction, inducing cell cycle arrest, apoptosis, and autophagy.

## 2. Results

### 2.1. Smp24 Inhibits the Proliferation of Hepatoma Cells

The cytotoxicity of Smp24 against both human hepatoma cell HepG2 and normal hepatic cell LO2 was evaluated. While compared with the minor effects on LO2 cells, Smp24 presented a remarkable inhibitory effect on the proliferation of HepG2 (Figure 1A). In detail, after 24 h incubation, the IC_50_ values of Smp24 against HepG2 and LO2 cells were approximately 5.524 µM and 16.68 μM, respectively. Moreover, as exhibited in Figure 1B, Smp24 decreased the viability of HepG2 hepatoma cells in concentration- and time-dependent manners. Consistently, in comparison with control group, the fluorescence of EdU-labeled cells was gradually decreased with the increasing Smp24 concentrations (Figure 1C). The morphological changes of HepG2 cells were visible after 24 h incubation with Smp24 (Figure 1D). Cells treated with Smp24 changed to a spherical shape with the appearance of cellular debris and floating cells, while it had an epithelial-like morphology with smooth surfaces in the control group. All these data indicated that Smp24 suppresses the viability of HepG2 cells in a dosage-dependent manner.

### 2.2. Smp24 Enters into HepG2 Cells through Endocytosis and Pore Formation

The surface charge was calculated due to the fact that the interaction between cancer cells and cationic peptide plays a crucial role in its inhibitory effect. As presented in Figure 2A, the zeta potential of HepG2 cells incubated with Smp24 (2.5, 5, and 10 μM) was increased from −14.67 mV to −8.11 mV in a concentration-dependent manner. Therefore, Smp24 could react with the HepG2 cell surface or cross the cell membrane, subsequently culminating in alterations of cell surface charge. To further evaluate the potential internalization activity of Smp24, the fluorescence of FITC-labeled Smp24 was observed. Smp24 was able to enter the HepG2 cells after 6 h of incubation, followed by the penetration into the nucleus in 24 h, as presented by the increase in fluorescence while compared with the control group (Figure 2B). Moreover, flow cytometry suggested that Smp24 also dose-dependently accomplished its internalization activity (Figure 2C). As a crucial component involved in the interaction between membrane and cell-penetrating peptide [12], the impact of heparan sulfate on the cellular internalization of Smp24 was assessed. Pretreatment with heparan sulfate (5, 10, and 20 μg/mL) led to the decreasing fluorescence of FITC-labeled Smp24 by approximately 1.56%, 48.12%, and 66.82%, respectively. Furthermore, treatment with ammonium chloride significantly reduced the cellular uptake of Smp24. The internalization effect of Smp24 in different temperatures was further analyzed. The cellular internalization of Smp24 at 37 °C was approximately 45.89% higher than that at 4 °C, showing that Smp24 penetrated HepG2 cell membrane in an energy-dependent and thermosensitive manner. These findings suggested the internalization of Smp24 into HepG2 cells via pore formation and endocytosis.

### 2.3. Smp24 Changes Cell Cytoskeleton Conformation

Actin-forming microfilaments in the cytoskeleton are involved in many cellular processes, such as cell signaling, division, motility, cytokinesis, the building-up and maintenance of cell junctions, and shape [13]. Accordingly, filamentous (F)-actin was stained with rhodamine–phalloidin to identify the effect of Smp24 on the cytoskeleton. In comparison with the control cells, which had flat microfilament bundles and sharp boundaries, treatment with Smp24 resulted in the reorganization of intracellular actin filaments together with the demolition of F-actin, leading to the structural dissociation of HepG2 cells. Furthermore, this phenomenon was aggravated by increasing Smp24 concentration (Figure 3). The results revealed the effect of Smp24 on the cytoskeleton in HepG2 cells by affecting the F-actin network.

### 2.4. Smp24 Destroys the Cellular and Mitochondrial Membrane of HepG2 Cells

The LDH-release assay is a common cell death/cytotoxicity assay used to assess plasma membrane damage to a cell population. As shown in Figure 4A, LDH release from HepG2 cells was increased with treatment of Smp24 in concentration- and time- dependent manners. SEM imaging was carried out to confirm the impact of Smp24 on the cell membrane of HepG2. Compared with the well-defined cell membrane of control cells, Smp24 treatment led to visible changes in morphology, with unclear boundaries of cell membrane together with leakage of cellular contents (Figure 4B). Consequently, the leaked calcein from HepG2 cells was measured to evaluate the impact of Smp24 on cell membrane permeability. As showed by the changing fluorescence intensity in Figure 2C, the calcein leakage from HepG2 cells was gradually increased by the administration of Smp24. Thus, the membrane integrity of HepG2 cells could be disrupted by Smp24. Furtherly, CoCl_2_ is a calcein fluorescence quencher in cytoplasm rather than in mitochondria. The calcein leakage fluorescence in the presence of CoCl_2_ was concentration-dependently decreased by Smp24 in comparison with the control group. Additionally, there was no statistically noteworthy change in the fluorescence intensity of cells incubated with both NAC and CsA. These findings indicate that Smp24 affects both the mitochondrial and cellular membranes of HepG2 cells in a nonspecific manner.

### 2.5. Smp24 Declines Mitochondrial Membrane Potential but Promotes ROS Production

The disruption of the mitochondrial membrane may result in the dysfunction of mitochondria, loss of mitochondrial membrane potential, as well as a high level of ROS [14], which finally cause metabolic cell death. Thus, the changes in membrane potential of Smp24-treated cells were assessed with JC-1 staining, which formed red fluorescence at a high mitochondrial membrane potential while forming green fluorescence at a low one. In comparison with the control group, Smp24 concentration-dependently increased the green fluorescence while declining the red fluorescence (Figure 5A), indicating that Smp24 could decrease the mitochondrial membrane potential.

DCFH-DA can be hydrolyzed by intracellular esterase to produce DCFH which does not permeate cell membranes and can be oxidized to generate fluorescent DCF. Consistent with the result in the JC-1 staining assay, Smp24 dose-dependently induced ROS production in HepG2 cells (Figure 5B), compared with the control cells. In addition, co-treatment with NAC notably ameliorated all the abnormal phenomena caused by Smp24. All these findings revealed that Smp24 is responsible for the loss of mitochondrial membrane potential and increase in ROS production, implying a damaged mitochondrial membrane and function. 

### 2.6. Smp24 Activates Mitochondrion-Mediated Intrinsic Pathway and Induces Apoptosis in HepG2 Cells

The variations in mitochondrial outer membrane permeability contribute to the apoptotic cascade in cell death pathways [15]. Thus, HepG2 cell apoptosis induced by Smp24 was investigated using DAPI and annexin V-FITC/PI staining assays. As presented in Figure 6A, typical apoptotic features, such as chromatin condensation and the appearance of apoptotic bodies, were obvious in HepG2 cells incubated with Smp24. The apoptotic cell rate measured using an annexin V-FITC/PI staining assay was also significantly increased from 8.64% to 46.8% after exposure to Smp24 (Figure 6B). Additionally, co-treatment with antioxidant NAC remarkably reduced the pro-apoptotic effect of Smp24, in which the apoptotic cell proportion decreased from 28.76% to 18.80%.

Due to their important role in the process of apoptosis, the mitochondria/cytochrome C mediated apoptotic pathways were examined to further explore the underlining mechanism of the apoptosis-induced activity of Smp24 in HepG2 cells [16,17]. Under our condition, treatment with Smp24 concentration-dependently upregulated the expression of cytochrome *C* (Figure 6C,D). What is more, Smp24 significantly increased the levels of cleaved caspase-3, caspase-9, and PARP, while its precursors were declined in a concentration-dependent manner. Simultaneously, compared with normal cells, Smp24-treated HepG2 cells possessed a remarkable increase in the expression of Bax while possessing a significant decrease in that of Bcl-1.

### 2.7. Smp24 Regulates the Expression of G2/M Phase-Related Proteins and Induces Cell Cycle Arrest

Mitochondrial damage is known to be related to the occurrence of cell-cycle accumulation, leading to apoptosis in cancer cells [18]. Therefore, flow cytometry was conducted to investigate the cell cycle change induced by Smp24 in HepG2 cells. After 24 h of treatment, the proportion of HepG2 cells arrested in the S phase was increased by Smp24 (2.5, 5, and 10 µM) from 12.87% to approximately 15.95%, 18.85%, and 20.85%, respectively, while those in the G2/M phase were also raised from 6.83% to 7.83%, 9.83%, and 14.83%. Moreover, the proportion of cells in phase G0/G1 was obviously dropped by Smp24 in a concentration-dependent manner (Figure 7A). In line with the changes in cell cycle, treatment with Smp24 upregulated the expression of cyclin A, cyclin B, p21^Wafl/Cip1^, as well as p53, while dose-dependently depressed the expression of CDK2 and cyclin E (Figure 7B,C).

### 2.8. Smp24 Regulates Autophagy-Related Signaling Pathways

Autophagy is a catabolic pathway essential for organismal homeostasis; together with apoptosis, it might be considered as a potential cancer cellular control [19,20]. Hence, the effect of Smp24 on autophagy-regulated signaling pathways was explored. As displayed in Figure 8A, when compared to the control cells, Smp24 markedly reduced the contents of phosphorylated mTOR, PI3K, as well as AKT, in a dose-dependent manner. Furthermore, treatment with Smp24 dose-dependently suppressed the expression of microtubule-associated protein LC3A/B-I and p62 while enhancing the levels of autophagosome formation marker LC3A/B-II [21].

The MAPK pathway may become a promising target in cancer therapy owing to the important role in multiple cell process, especially in autophagy, apoptosis, and cell cycle arrest [22,23]. Under our conditions, Smp24 significantly decreased the expression of phosphorylated-JNK, ERK, and p38, while it did not affect its total protein expression (Figure 8C,D). These results suggested that Smp24 induced autophagy in HepG2 cells by regulating autophagy-regulated signaling pathways.

### 2.9. Smp24 Shows Antitumor Effects In Vivo

The in vivo antihepatoma effect of Smp24 was evaluated in HepG2 xenograft mice. As presented in Figure 9B–D, Smp24 significantly reduced tumors in both weight and volume by approximately 55.4% and 56.3%, respectively, in comparison with the control group at the end of experiment. Furthermore, Smp24 did not cause notable changes in body weight and crucial organs including liver, heart, lung, spleen, and kidney in the xenograft mice (Figure 9E,F). Consistent with the above results, histopathological examination indicated the antitumor effect of Smp24 by the appearance of abnormal changes in cancer cells, such as nuclear condensation, cell shrinkage, and inflammatory cell infiltration (Figure 9G). Furthermore, in line with the upregulated expression induced by Smp24 in vitro, the expression of cleaved caspase-3 was also verified in tumor tissue by immunohistochemical staining assay (Figure 9H). These findings reveal the antitumor effect of Smp24 in vivo.

## 3. Discussion

Despite their multidrug resistance and toxicity towards normal tissue, radiotherapy and chemotherapy remain the most common non-surgical remedies in cancer treatment. However, new treatments with less side effects are urgently needed. AMPs become promising anticancer agents due to their targeting of cell membranes instead of specific receptors, making them less likely to induce drug resistance in tumor cells [24]. The cytotoxicity of Smp24 against HepG2 cells have been reported in a previous study [8]; however, its underlying mechanism remains elusive. In the present study, the antitumor activity of Smp24 against HepG2 has been confirmed both in vitro and in vivo. We also revealed that Smp24 significantly suppresses the proliferation of HepG2 cells, while having a slight effect on that of the normal hepatic cell LO2, suggesting a highly selective activity against tumor cells (Figure 1A). It is noteworthy that the IC_50_ values of Smp24 against HepG2 cells was slightly higher than the results in our previous study against A549 cells [10,11]. Therefore, there are some common mechanisms against two kinds of tumor by Smp24.

Cationic AMPs commonly possess cancer-selective toxicity due to the fact that fundamental differences exist in cell membranes between normal cells and cancer cells [25]. The normal mammalian cell membrane outer leaflet is primarily composed of neutral zwitterionic phospholipids, whereas the cancer cell membrane consists of exclusive anionic constituents such as O-glycosylated mucins [26], phosphatidylserine [27], sialylated gangliosides [28], and heparan sulfate [29]. The presence of positively charged residues in AMPs enhances electrostatic binding to negative charges induced by anionic constituents in cancer cell membranes, causing changes in surface charge [30]. In agreement, Smp24 enhances the zeta potential of HepG2 cells, while its cellular internalization is suppressed in the presence of heparan sulfate (Figure 2A,C).

Antitumor AMPs generally possess a consistent membranolytic mode of action as pore formation, which cause cancer cell membrane disruption or permeation, leading to cell lysis and death [31,32]. In this study, evidence from LDH staining, SEM analysis, and calcein AM assay has revealed that Smp24 is likely a pore-forming peptide due to the occurrence of disruption and increasing permeability of cancer cell membranes (Figure 2 and Figure 4). However, further studies are required to observe the pore formed by Smp24. Additionally, the responsibility of endocytosis for the uptake of Smp24 by HepG2 cells is also demonstrated, which is similar with melittin, a cationic AMP that enters cancer cells via the endocytosis-dependent pathway [33].

The membranolytic effect of AMPs is not only defined on the cell membrane, but also on the mitochondrial membrane, causing the decrease in its potential, altering the permeability of the mitochondrial membrane, increasing ROS production, and eventually mitochondrial dysfunction [5]. For instance, myristol-CM4, a new synthetic analog of AMP CM4, exerts its antitumor capacity against breast cancer cells by targeting mitochondria, which leads to a release of pro-apoptotic factors such as cytochrome *C* and induces mitochondria-dependent apoptosis [34]. Consistently, the damaged mitochondrial membrane caused by Smp24 was presented by the decline in its potential, and an increase in ROS accumulation, as well as calcein leakage from mitochondria (Figure 4 and Figure 5). Consequently, Smp24 enhances apoptosis in HepG2 cells via activating the intrinsic mitochondrial apoptotic pathway (Figure 6). These data reveal that the antitumor mechanism of Smp24 is attributable to target mitochondria and induce mitochondria-mediated apoptosis.

The cytoskeleton is known to contribute to cancer progress by inducing cell proliferation and activating oncogenes, leading to tumorigenesis [35]. As an important component in cytoskeletal structures, the actin skeleton is involved in multiple mitochondrial functions, including mitochondrial dynamics, trafficking and autophagy, mitochondrial biogenesis, and metabolism [36]. Notably, the disruption of actin dynamics might cause the decrease in mitochondrial membrane potential, increase the ROS accumulation, induce the intrinsic apoptosis pathway, and finally lead to cell death [37]. In line with this, the reorganization of F-actin in the HepG2 cell cytoskeleton was observed in the presence of Smp24 (Figure 3). However, contrary to its effects on A549 cells [10,11], Smp24 could not inhibit the mobility of HepG2 cells in the wound-healing and transwell assays (data not shown). Therefore, there are some discrepancies in its mechanism against two cell lines and further research is necessary.

Mitochondria are crucial in cellular metabolism and their dysfunction might affect cell proliferation, causing cell cycle accumulation and apoptosis [38]. In agreement, mitochondria damage caused by Smp24 causes cell cycle suspension in the S and G2/M phases in HepG2 cells, as presented in flow cytometry analysis (Figure 7). p53 is a transcription factor which exerts a critical effect both in the G1/S and G2/M checkpoint of cell cycle and can be regulated by the transcriptional activation of p21^Waft1/Clip1^ [39,40]. Concurrently, p21 ^Waft1/Clip1^ also causes G1/S phase cell cycle arrest by suppressing the kinase activity of CDK2/cyclin A, CDK2/cyclin E, and CDK1/cyclin A, while the inhibition of G2/M transition is related to the binding with CDK1/cyclin B1 [41]. Consistently, Western blotting reveals that treatment with Smp24 dose-dependently suppresses the expression of cyclin E and CDK2, but upregulates the expression of cyclin A, cyclin B, p21^Waft1/Clip1^, and p53, inducing cell cycle accumulation at the S and G2/M phases (Figure 7).

Both autophagy and apoptosis are of pivotal importance in cellular homeostasis, and there is abundant evidence proving the co-regulation of these pathways. For example, overexpression of the autophagy effector Beclin 1 may lead to the release of Bak/Bax from Bcl-2 to enhance apoptosis, whereas an excessive Beclin 1-dependent autophagy occurs in the absence of Bcl-2 [42,43]. Several peptides have been identified to possess antitumor activity through both the autophagy and apoptosis pathways, such as CTLEW [44] and FK-16 [45]. Furthermore, the PI3K/Akt/mTOR pathways are regarded as promising targets for cancer therapy due to their important role in the process of cell proliferation, growth, survival, and mobility [46]. Consequently, multiple candidate anticancer agents suppress tumor growth by inhibiting PI3K/Akt/mTOR pathways and then inducing autophagy, as well as apoptosis of cancer cells [47,48]. In line with these, Smp24 suppressed the phosphorylation of PI3K/Akt/mTOR and increased the autophagic flux of LC3A/B-II/I, as well as the degradation of p62 (Figure 8A,B). It also is acknowledged that the MAPK signaling pathway is associated with the autophagy and apoptosis of cancer cells [22,23,49] and some compounds suppressing both the PI3K/Akt/mTOR and MAPK pathways induce autophagy and apoptosis of hepatocellular carcinoma cells [50]. Similarly, Smp24 suppresses the MAPK signaling pathways (Figure 8C,D). It is also noteworthy that Smp43, another AMP derived from *Scorpio*
*Maurus palmatus*, and quercetin, a member of the flavonoid family, induce autophagy and apoptosis by activation of the MAPK signaling pathway [51,52]. The discrepancy in the mechanism might be associated with the cell response, which relies on the nature, strength, and duration of the MAPK pathways activated [53]. Thus, further research is needed to explore the detailed mechanism of how Smp24 induces both apoptosis and autophagy by inhibiting MAPK signaling pathways.

Despite the effectiveness and selectivity against tumor cells in vitro experiments, the use of peptide-based anticancer agents in clinical trials is still a challenge due to the degradation by proteases in vivo [54]. Our animal experiments revealed that 2 mg/kg Smp24 treatment leads to a significant decrease in tumor weight and volume, while it does not cause notable changes in body weight and important organs, which is consistent with the previous study [10,11], indicating the considerable stability and high selectivity of Smp24 against tumor tissues in vivo (Figure 9). Thus, Smp24 is an ideal antitumor candidate molecule.

## 4. Conclusions

In summary, the present study reveals the underlying mechanism of Smp24 against HepG2 cell proliferation. Smp24 enters HepG2 cells via pore formation and endocytosis, resulting in mitochondrial dysfunctions and membrane defects, consequently causing cell necrosis, cycle arrest, apoptosis, and autophagy. The anti-hepatoma activity is also verified in xenograft mice. Thus, our results provide evidence of the antitumor mechanisms of AMPs and suggest that Smp24 might be a promising candidate in hepatocellular carcinoma therapy.

## 5. Materials and Methods

### 5.1. Animals and Ethics Statement

Six-week-old BALB/c nude mice (18–20 g) were bought from the Laboratory Animal Center of Southern Medical University. Mice were randomly divided into the control and Smp24 (2 mg/kg)-treated groups and were separately housed in groups of five in a SPF mini-barrier system at the central animal facility of Southern Medical University. Animals lived under controlled 21 ± 2 °C room temperature, 60% humidity with a 12 h light-dark cycle. All experiments with animals were given the approval by the Animal Ethics Committee of Southern Medical University (Guangzhou, China) with ethical approval number: L2019226.

### 5.2. Chemicals and Cell Culture

Phosphate-buffered saline (PBS), fetal bovine serum (FBS), Dulbecco’s modified Eagle’s medium (DMEM), and trypsin were all bought from Gibco (Grand Island, NY, USA). HepG2 and LO2 cell lines were purchased from the American Type Culture Collection (Manassas, VA, USA). Cells were grown in RPMI-1640 medium including 1% penicillin-streptomycin and 10% FBS in a 37 °C incubator supplied with 5% CO_2_. cytochrome C, Bax, Bcl-2, p53, p21, cleaved PARP, PARP, cleaved caspase-3, caspase-3, cleaved caspase-9, caspase-9, CDK2, cyclin A, cyclin B, cyclin E, p-JNK, JNK, p-Erk, Erk, p-p38, p38, p-Akt, Akt, p-mTOR, mTOR, p-FAK, FAK, p-PI3K, PI3K, LC3A/B, p62, GAPDH, β-actin, and all secondary antibodies were purchased from Cell Signaling Technology (Beverly, MA, USA). DAPI, cyclosporin A (CsA, an inhibitor of permeability transition), N-acetyl-L-cysteine (NAC), ROS assay kit, LDH-release assay kit, cell cycle and apoptosis analysis kit, and mitochondrial membrane potential assay kit for JC-1 were bought from Beyotime Institute of Biotechnology (Shanghai, China). Smp24 and FITC-labeled Smp24 were obtained as previously described by us [9].

### 5.3. Cell Viability and Proliferation Analysis

Effect of Smp24 on cellular viability was determined using MTT assays as previously described by us [10,11]. In brief, HepG2 and LO2 cells (1 × 10^4^ cells/well) were exposed to a sequence of concentrations of Smp24 (1.25–20 μM) in 96-well plates for 12, 24, and 48 h, respectively. After that, cells were incubated with 10 µL MTT (5 mg/mL) at 37 °C for 4 h in the dark. The cell medium was subsequently discarded and replaced by 200 µL DMSO before the determination of optical density was conducted using a microplate reader (Tecan Company, Männedorf, Switzerland) at an absorbance of 490 nm. Considering that the MTT assay may be biased, with disruption of mitochondrial dehydrogenase system resulting in mitochondrial dysfunction or apoptosis, the cell proliferation was measured with the BeyoClick™ EdU cell proliferation kit with Alexa Fluor 488 (Beyotime Institute of Biotechnology, Shanghai, China) in accordance with the manufacturer’s manual. Briefly, HepG2 cells (2 × 10^5^ cells/well) were treated with a sequence of concentrations of Smp24 (0, 1.25, 2.5, 5, and 10 μM) in 6-well plates for 24 h. Subsequently, the cells were treated with EdU for 2 h, followed by staining with Alexa Fluor 488 under protection from light for 30 min. The fluorescence intensity was analyzed using flow cytometry (Becton Dickinson Company, Bedford, MA, USA) with 495 nm excitation and 519 nm mission wavelengths. The experiments were repeated in triplicate.

### 5.4. Peptide Internalization Measurement

Flow cytometry and fluorescence microscope (Axio Observer, Zeiss, Oberkochen, Germany) were used to measure the internalization of FITC-labeled Smp24. In brief, HepG2 cells (1 × 10^5^ cells/well) were grown overnight on 24-well plate and then co-incubated with 2.5, 5, and 10 μM of FITC-labeled Smp24 for 1 h and 6 h at 37 °C. After washing with PBS, the fluorescence intensity was determined with flow cytometry. For the location analysis of Smp24 in cells, after being treated with 5 μM FITC-labeled Smp24 at 37 °C for different times (6, 12, and 24 h), the cells were washed with PBS, fixed with 4% paraformaldehyde (PFA) for 30 min, stained with DAPI for 10 min, and finally observed under fluorescence microscope at 400× magnification. Approximately three single-plane images each well were captured. 

To ascertain whether heparan sulfate affects the internalization of Smp24, FITC-labeled Smp24 at 5 µM was mixed with heparan sulfate at 5, 10, or 20 µg/mL in RPMI-1640 medium for 30 min, and then were incubated together with HepG2 cells at the density of 1 × 10^5^ cells/well for 1 h. The cell fluorescence intensity was measured with flow cytometry.

To ascertain whether the internalization of Smp24 depends on the cellular energy state, HepG2 cells were pre-incubated for 30 min at 37 °C and 4 °C, and then with 5 µM FITC-labeled Smp24 for another 1 h. After that, the cells were washed with PBS and measured with flow cytometry. In another set of experiments to further confirm the effects of the cellular energy state on its internalization, HepG2 cells were pre-incubated with 50 mM NH_4_Cl for 30 min before being treated with 5 µM FITC-labeled Smp24 for another 1 h and then measured with the flow cytometry. All experiments were repeated at least three times.

### 5.5. Membrane Integrity Assay

Calcein AM staining was conducted to evaluate the membrane integrity of HepG2 cells in accordance with the manufacturer’s manual (Beyotime Institute of Biotechnology, Shanghai, China). In short, HepG2 cells (1 × 10^5^ cells/well) were grown overnight in a 12-well plate and subsequently incubated with a sequence of concentrations of Smp24 for 24 h. Cells were then digested by trypsin, washed with PBS for three times, and treated with 1 µM calcein AM for 30 min at 37 °C. The cells were finally analyzed with flow cytometry to detect the levels of calcein AM after being washed again with PBS. In another set of experiments, cells were pre-incubated with 1 µM CsA before treatment with calcein AM or were incubated with CoCl_2_ after treatment with calcein AM to determine the integrity of mitochondrial membrane. Zeta potential analysis was carried out to further confirm the membrane integrity. In short, 1 × 10^5^ HepG2 cells were re-suspended in PBS and mixed with Smp24 (0, 2.5, 5, 10 µM) for 10 min at room temperature. Folded capillary cell (DTS1070) and Zetasizer system (Nano ZS; Malvern Instruments Ltd., Worcestershire, UK) were applied to measure zeta (ξ) potential of the aboveHepG2 cells. All experiments were repeated at least three times.

### 5.6. Cell Morphology Observation

In the present study, 2 × 10^5^ HepG2 cells were cultivated on 6-well plate overnight and then exposed to a sequence of concentrations of Smp24 (0–20 μM) for 24 h. The cellular morphology was then observed with an inverted phase contrast microscope (100× magnification). Approximately 3 images each well were captured.

### 5.7. LDH-Release Assay

The LDH assay was accomplished in accordance with the manufacturer’s manual. Briefly, after incubation with a gradient of concentrations of Smp24 (0–20 μM) for 12, 24, and 48 h, respectively, each well containing HepG2 cells was supplemented with 10 µL of LDH-release solution and incubated for 1 h. After being added to a new plate, the supernatants in each well were then incubated with 60 µL of substrate solution in the dark for 30 min. A microplate reader (Tecan Company, Männedorf, Switzerland) was used to determine the absorbance value of the mixture at 490 nm. The LDH release rate (%) = (absorbance of sample − absorbance of control)/(absorbance of max LDH activity) × 100. All experiments were carried out at least three times.

### 5.8. Scanning Electron Microscopy Determination

After 24 h of growth in a 12-well plate on the glass coverslips, HepG2 cells (1.2 × 10^5^ cells/well) were co-cultured with a gradient of concentrations of Smp24 (0–10 μM) for another 24 h with PBS as a negative control. Subsequently, the cells were fixed with 4% glutaric dialdehyde and 2.5% glutaric dialdehyde at 22 °C for 2 h and 4 °C for 8 h, respectively. A series of gradient ethanol/water solutions was then used to dehydrate the samples. After that, cells were coated by gold and observed with Phenom ProX instrument (Phenom World, Eindhoven, The Netherlands) at 15 kV. The experiments were detected in triplicate.

### 5.9. Fluorescence Microscopy Analysis

Fluorescence microscopy analysis was used to observe the F-actin reorganization, intracellular ROS accumulation, and mitochondrial membrane potential. To observe the changes in F-actin, after overnight incubation in 24-well plates, HepG2 cells (7 × 10^4^ cells/well) were cultivated with Smp24 at the concentration of 0–10 μM for another 24 h. After that, cells were fixed with 4% PFA, dyed with rhodamine–phalloidin for 30 min, washed with PBS for three times, and dyed again with DAPI for another 10 min. Subsequently, fluorescence microscope at 400× magnification was used to observe the changes in F-actin.

To determine intracellular ROS levels, HepG2 cells were co-cultured with Smp24 (0, 2.5, 5, and 10 μM) for 12 h and then stained with 10 μM 2,7-dichlorodihydro-fluoresceindiacetate (DCFH-DA) for 30 min at 37 °C under protection from light. The supernatant was removed, and cells were washed three times with serum-free cell culture medium to adequately remove DCFH-DA that did not enter the cells. An inverted fluorescence microscopy at 200× magnification was applied to clarify the changes in cells. NAC-treated cells were considered as a positive control.

To detect the impact of Smp24 on the mitochondrial membrane potential, HepG2 cells were co-cultured with a gradient of concentrations of Smp24 for 12 h and then were dyed with JC-1 at 37 °C for 30 min in accordance with the kit instruction manual. The supernatant was removed, and cells were washed twice with JC-1 staining buffer (1×) and supplemented with 2 mL of cell culture medium, which may contain serum and phenol red. The changing mitochondrial membrane potential was observed using fluorescence microscopy at 400× magnification, with NAC as a positive control. Approximately 3 random images each well were obtained.

### 5.10. Cell Cycle and Apoptosis Measurement

To define whether Smp24 affects cell cycle and apoptosis in HepG2 cells, flow cytometry was used. HepG2 cells (2 × 10^5^ cells/well) were seeded in 6-well plates overnight and then co-cultured with a gradient of concentrations of Smp24 or 2 mM NAC for 24 h. After that, the cells were harvested with trypsin digestion, washed with cold PBS three times, fixed with 70% ethanol at 4 °C for 12 h, and finally dyed with PI for 30 min at 37 °C to measure cell cycle. For the apoptosis measurement, the cells were dyed with annexin V-FITC/PI instead of only PI at 22 °C for 15 min before being analyzed with flow cytometry. All experiments were conducted in triplicate.

### 5.11. Western Blot Analysis

HepG2 cells (2 × 10^5^ cells/well) were incubated with Smp24 (0, 2.5, 5, and 10 μM) in 6-well plates for 24 h. After that, the cells were harvested by trypsin digestion and then lysed with RIPA lysis solution including 1% protease and phosphatase inhibitors (FDbio Science Biotech Co., Ltd., Hangzhou, China) at 4 °C for 15 min. Subsequently, the supernatant was separated by SDS-PAGE and then transferred onto PVDF membrane (Millipore, Billerica, MA, USA). After being incubated with right primary antibodies (1:1000) for 12 h at 4 °C and then with corresponding secondary antibodies (1:2000) for 1 h at 25 °C, a hypersensitive ECL chemiluminescence agent was used to visualize blot bands which were analyzed with Image J software (64-bit, National Institutes of Health, MD, USA) (*n* = 3 replicates).

### 5.12. Animal Experiments

The right flank of each mouse was subcutaneously injected with HepG2 cells (5 × 10^6^ cells/mouse) with viability over 90%. The changes in tumor size were examined every day by palpation and determined using caliper in double planes. The volume of tumors was calculated according to the following equation: volume (mm^3^) = (smallest diameter)^2^ × (largest diameter)/2. Once the volume of tumors increased to approximately 50 mm^3^, physiological saline or Smp24 (2 mg/kg) was injected once every three days for a total of six times near the tumor site. Including the tumor volumes, the body weight of mice was also recorded daily. At the end of experiment, the tumors and important organs including liver, heart, lung, spleen, and kidney were separated, weighed, and photographed. After being embedded in paraffin, HE staining reagent was applied to assess the morphological changes of tumor cells in vivo. To evaluate the cell apoptosis levels of tumor tissue, immunohistochemical staining was carried out to examine the contents of cleaved caspase-3 in tumor tissues. The results were obtained from five mice each group.

### 5.13. Statistical Analysis

All data were expressed as mean ± SEM and analyzed using GraphPad Prism 5.0 (GraphPad Software, Inc., La Jolla, CA, USA) by one-way ANOVA with Bonferroni’s multiple comparison. Statistical significances were shown as * *p* < 0.05, ** *p* < 0.01, and *** *p* < 0.001.

## Figures and Tables

**Figure 1 toxins-14-00717-f001:**
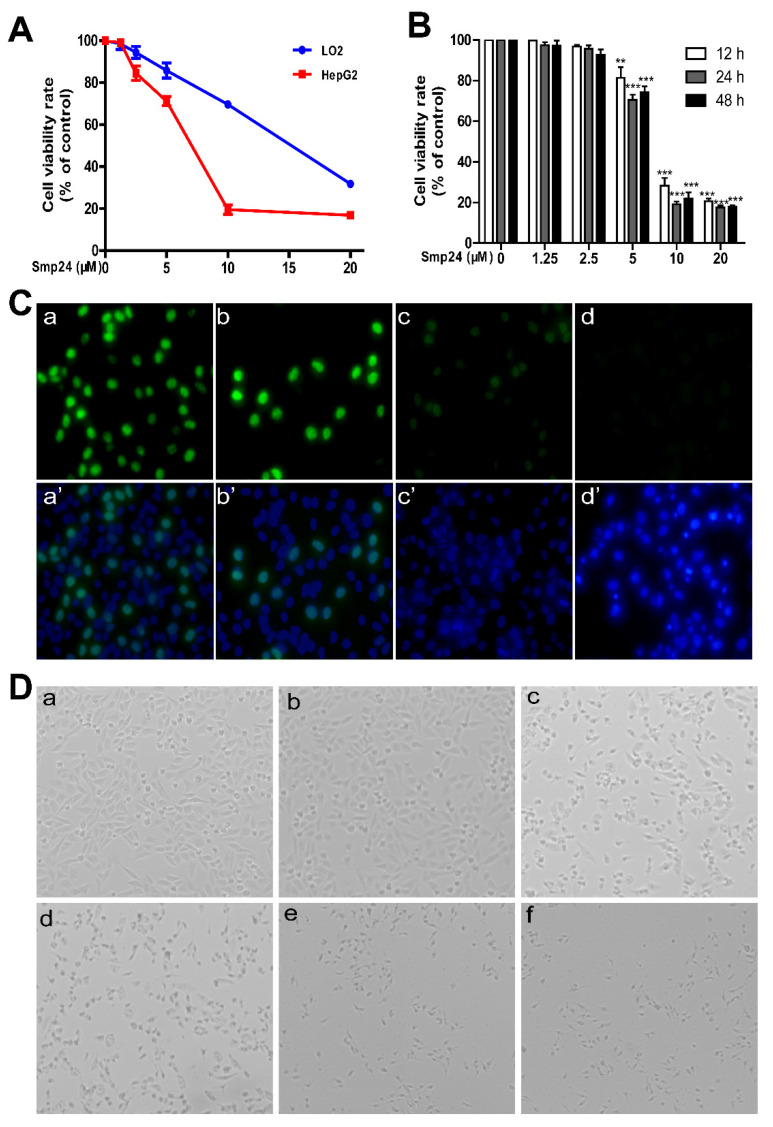
Smp24 suppressed the proliferation of HepG2 cells. (**A**) Effect of Smp24 on the viability of HepG2 and LO2 cells. After treatment with Smp24 (0–20 μM) for 24 h, MTT assays were used to determine cell viabilities. (**B**) Cytotoxicity of different concentrations of Smp24 against HepG2 cells after 12, 24, and 48 h of treatment and their IC_50_ values were detected by MTT assay. (**C**) EdU cell proliferation assay. Photographs were captured at 400× magnification. Panels a–d present cells stained with EdU while panels a’–d’ present the merge images of cells stained by EdU and DAPI. Panels a and a’: control cells; panels b–d and b’–d’: cells treated with 2.5, 5, and 10 μM Smp24 for 24 h, respectively. (**D**) Changes in cell morphology in the presence of Smp24. Photographs were taken at 100× magnification. Panel a: control group; panels b–f: cells treated with 1.25, 2.5, 5, 10, and 20 μM of Smp24 for 24 h, respectively. The data are shown as the mean ± SEM (*n* = 3). ** *p* < 0.01 and *** *p* < 0.001 are regarded to be statistically significant in comparison with the control group.

**Figure 2 toxins-14-00717-f002:**
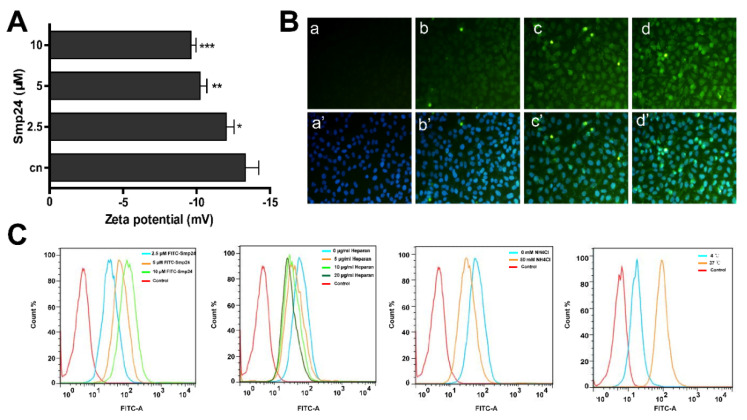
Uptake mechanism of Smp24 into HepG2 cells. (**A**) Zeta potential assay. Changes of cell membrane surface potential after Smp24 (0, 2.5, 5, and 10 µM) was co-incubated with HepG2 cells for 5 min with PBS as the control; (**B**) fluorescence microscope detection of Smp24 entering HepG2 cells. The upper panel depicts FITC-labeled Smp24-stained cells, the lower panel depicts cells stained by FITC-labeled Smp24 and DAPI, panels a and a’: the control groups, panels b–d and b’–d’: the 6, 12, and 24 h Smp24 administration groups, respectively; photographs were taken at 400× magnification. (**C**) Flow cytometry measurement of Smp24 entering HepG2 cells in the absence and presence of ammonium chloride, heparan sulfate (5, 10, and 20 μg/mL) or different temperatures for 1 h. The data are shown as the means ± SEM (*n* = 3). * *p* < 0.05, ** *p* < 0.01, and *** *p* < 0.001 are regarded to be statistically significant in comparison with control group without Smp24 treatment.

**Figure 3 toxins-14-00717-f003:**
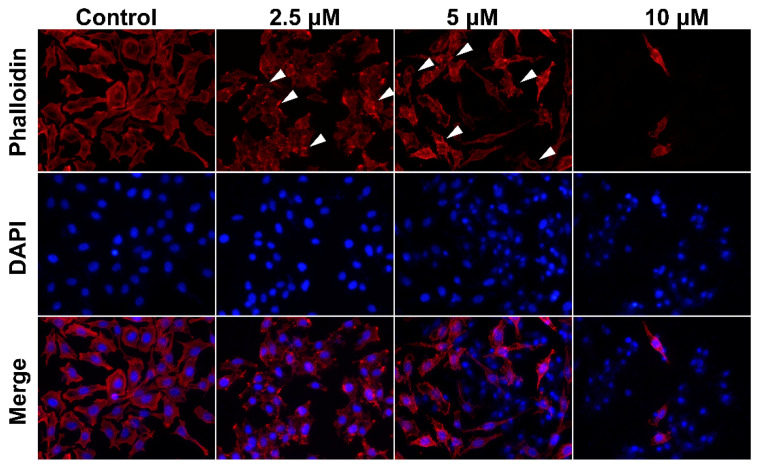
The changes in the cytoskeleton of HepG2 cells in the presence of Smp24. After being treated with Smp24 (2.5, 5, and 10 μM) for 24 h, HepG2 cells were subsequently dyed by rhodamine–phalloidin and DAPI and were examined under fluorescence microscopy with 400× magnification. White arrows indicate disorganized microfilament bundles. Scale bar, 20 μm.

**Figure 4 toxins-14-00717-f004:**
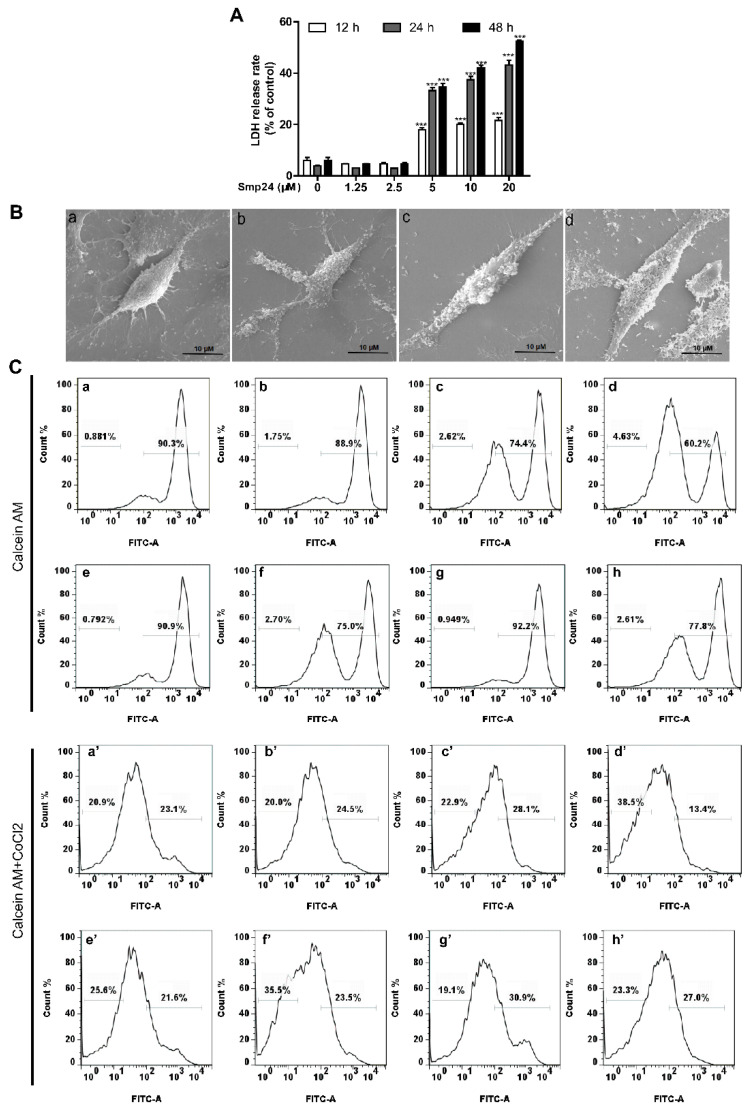
Cellular and mitochondrial membrane damage stimulated by Smp24 in HepG2 cells. (**A**) Levels of LDH release from Smp24-treated HepG2 cells for 12, 24, and 48 h. (**B**) The morphological structure changes of HepG2 cells under SEM. Panels a–d: HepG2 cells in the presence of 0, 2.5, 5, and 10 μM of Smp24, respectively. Scale bar: 10 μm. (**C**) Flow cytometry analysis of calcein fluorescence with (**top panels**) and without CoCl_2_ (**bottom panels**). After pre-incubation with NAC (2 mM) or CsA (1 μM), cells were subsequently treated by 5 μM Smp24 for 24 h, followed by incubation with 1 μM calcein AM or 1 μM calcein AM + 1 mM CoCl_2_ at 37 °C for 30 min. Panels a–d and Panels a’–d’: HepG2 cells sequentially treated with 0, 2.5, 5, and 10 μM of Smp24, respectively; panels e and e’: HepG2 cells treated by 2 mM NAC; panels f and f’: HepG2 cells treated by 2 mM NAC + 5 μM Smp24; panels g and g’: HepG2 cells treated by 1 μM CsA; panels h and h’: HepG2 cells treated by 1 μM CsA + 5 μM Smp24. The data are displayed as the means ± SEM (*n* = 3). *** *p* < 0.001 are considered statistically significant in comparison with the control group without peptide.

**Figure 5 toxins-14-00717-f005:**
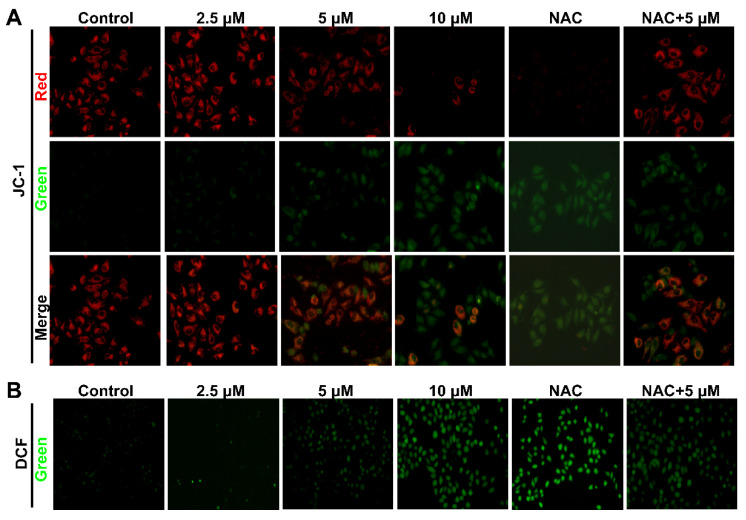
Impacts of Smp24 on mitochondrial membrane potential and ROS accumulation of HepG2 cells. (**A**) Mitochondrial membrane potential changes in HepG2 cells induced by Smp24. Cells were incubated with 0–10 μM Smp24, 2 mM NAC, or with 2 mM NAC + 5 μM Smp24 for 12 h, as described in Material and Methods, and then fluorescent images were taken under fluorescence microscope at 400× magnification. (**B**) ROS levels in HepG2 cells. HepG2 cells were incubated with 0–10 μM Smp24 for 12 h, followed by DCFH-DA staining at 37 °C for 30 min before detection by fluorescence microscope.

**Figure 6 toxins-14-00717-f006:**
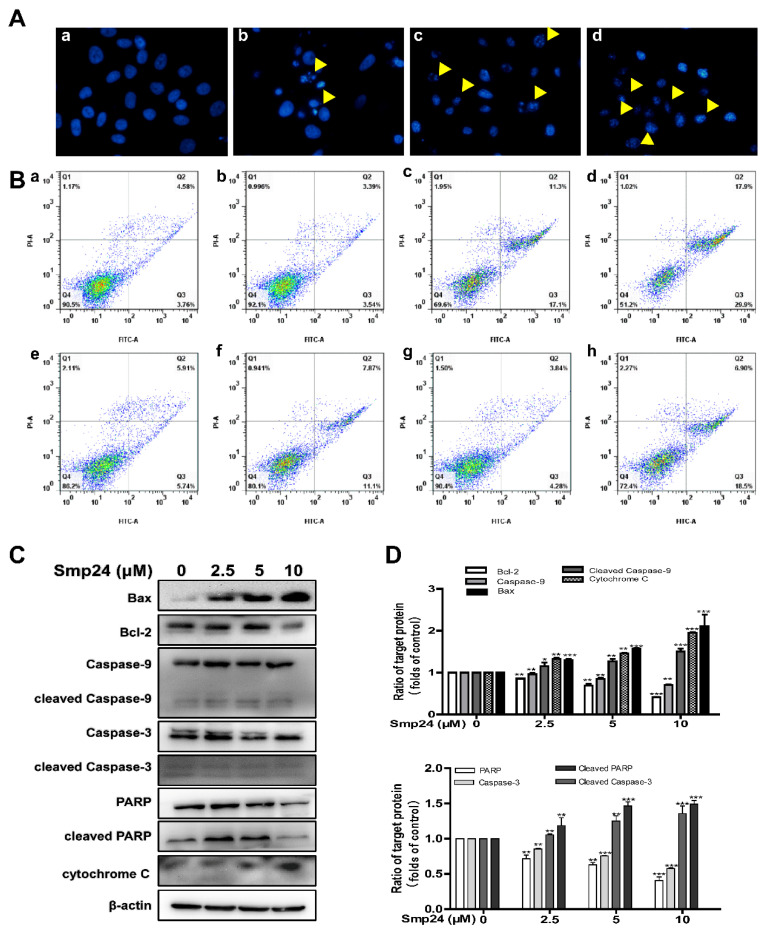
Apoptosis of HepG2 cells induced by Smp24. (**A**) DAPI staining images of apoptotic cells. HepG2 cells were incubated with 2.5, 5, and 10 μM Smp24 for 24 h, followed by DAPI staining. The apoptotic bodies were marked by yellow triangles. Panels a–d: HepG2 cells sequentially treated by 0, 2.5, 5, and 10 μM of Smp24. (**B**) Flow cytometry of apoptotic cells. HepG2 cells were incubated with 2.5, 5, and 10 μM of Smp24, and subsequently dyed with annexin V-FITC/PI. Panels a–d: HepG2 cells sequentially incubated with 0, 2.5, 5, and 10 μM of Smp24; panels e–f: HepG2 cells sequentially incubated with 2 mM NAC and 2 mM NAC + 5 μM Smp24; panels g–h: HepG2 cells sequentially incubated with 1 μM CsA and 1 μM CsA + 5 μM Smp24, respectively. (**C**) Representative Western blots of the apoptotic pathway induced by Smp24 in HepG2 cells. (**D**) Quantification analysis of band densities in (**C**). Bars mean the relative expression of the target protein to the control. Image J software (64-bit, National Institutes of Health, MD, USA) was applied to analyze the band, and the data are displayed as mean ± SEM (*n* = 3). ** *p* < 0.01 and *** *p* < 0.001 are regarded to be statistically significant in comparison with the control group.

**Figure 7 toxins-14-00717-f007:**
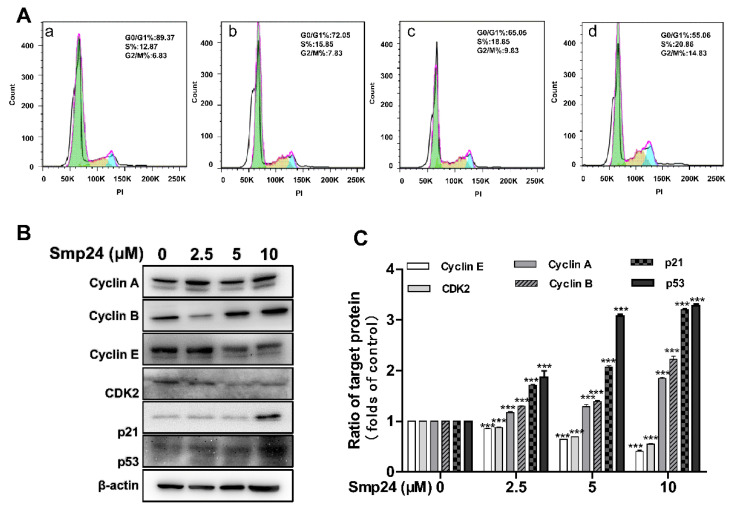
Cycle arrest of HepG2 cells induced by Smp24. (**A**) Cell cycle analysis by flow cytometry. Panels a–d: 0, 2.5, 5, and 10 μM Smp24-treated groups, respectively. (**B**) Western blot images of HepG2 cells treated by Smp24 (2.5, 5, and 10 μM) for 24 h. (**C**) Statistical analysis of (**B**). Bars mean the relative expression of the target protein to the control. Image J software was applied to analyze the band, and the data are shown as mean ± SEM (*n* = 3). *** *p* < 0.001 is regarded to be statistically significant in comparison with the control group.

**Figure 8 toxins-14-00717-f008:**
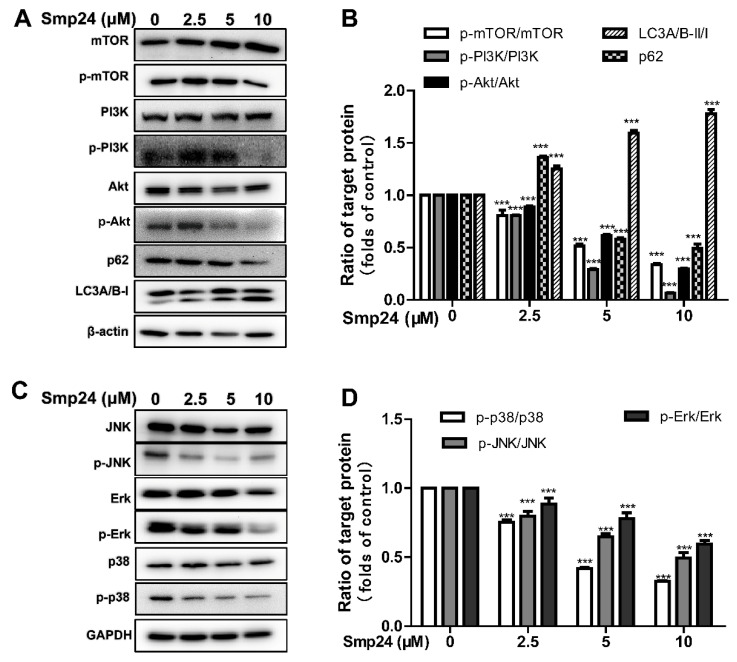
Autophagy and signaling pathways regulated by Smp24. (**A**,**C**) Representative Western blot images of proteins belonging to PI3K/Akt/mTOR and MAPK signaling pathways. HepG2 cells were exposed to 2.5, 5, and 10 μM Smp24 for 24 h, with PBS as the negative control. (**B**,**D**) Quantification analysis of band densities in (**A**,**C**). Bars mean the relative expression of the target protein to the control group. Data are shown as mean ± SEM (*n* = 3). *** *p* < 0.001 is regarded to be statistically significant in comparison with the control group.

**Figure 9 toxins-14-00717-f009:**
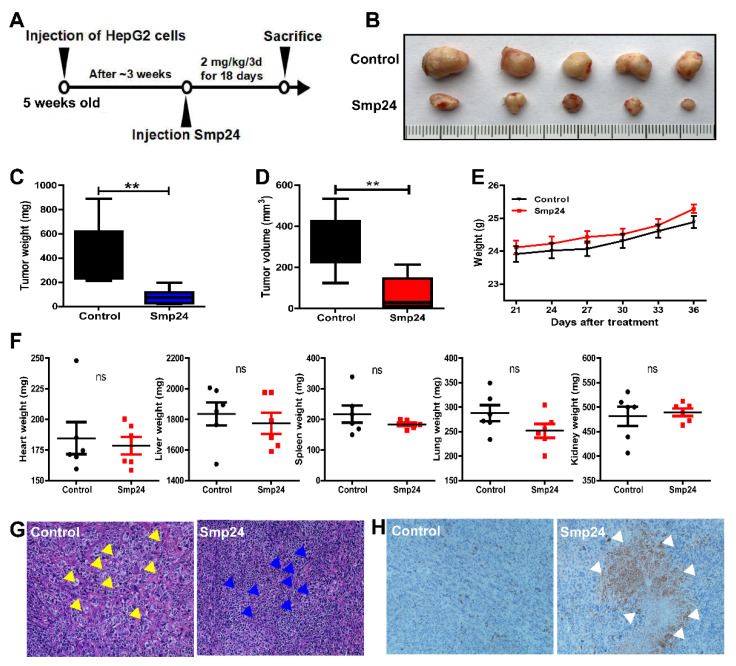
Effect of Smp24 on xenograft mice. Once the volume of tumors increased to approximately 50 mm^3^, physiological saline or Smp24 (2 mg/kg) was injected near the tumor site every three days for a total of six times. The tumor volumes and the body weight of mice was also recorded daily. The tumors and important organs including liver, heart, lung, spleen, and kidney at the end of experiment were separated for analysis. (**A**) Schedule for in vivo experiment. (**B**) Images of tumors removed from mice. Scale unit: centimeter. (**C**,**D**) Tumor weight and volume at the end of experiments. (**E**) Body weight changes of nude mice after Smp24 injection. (**F**) Organ weight changes at the end of experiments. (**G**) HE staining images of tumor tissue. (**H**) IHC images of cleaved caspase-3 in tumor tissue from xenograft mice. Photographs were taken under light microscope at 200× magnification. Yellow arrow: enlarged tumor cells. Blue arrow: shrinking cells. White arrow: positive apoptotic staining (brown areas). The data are expressed as mean ± SEM (*n* = 5). ns: no significance, ** *p* < 0.01 is regarded to be statistically significant in comparison with control group.

## Data Availability

The datasets used and analyzed during the current study are available from the corresponding author on reasonable request.

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
