# Peer review of "Smp24, a Scorpion-Venom Peptide, Exhibits Potent Antitumor Effects against Hepatoma HepG2 Cells via Multi-Mechanisms In Vivo and In Vitro"

_toxins, 2022, doi:10.3390/toxins14100717_

Round 1
Reviewer 1 Report
The reviewer would like to thank the authors for their work that hypothesizes that the scorpion-derived peptide Smp24 kills cancer cells of the HepG2 cell line in an animal model with, presumably, minimal off-target effects. The reviewer would like to provide a critique of the article from a stylistic and technical terms.
Style:
1. The reviewer notes that the article is a near identical copy to the following paper:
Guo R, Liu J, Chai J, Gao Y, Abdel-Rahman MA, Xu X. Scorpion Peptide Smp24 Exhibits a Potent Antitumor Effect on Human Lung Cancer Cells by Damaging the Membrane and Cytoskeleton In Vivo and In Vitro. Toxins (Basel). 2022 Jun 28;14(7):438. doi: 10.3390/toxins14070438. PMID: 35878176; PMCID: PMC9318729.
The experiments seem to be very similar albeit with a different main cell line of focus.
The work presented seems to be novel, however, given that HepG2 is a liver cell line and A594 cells are derived from lung tissue.
The authors, should, at very least, cite and discuss this paper given that there are many similar conclusions in both.
2. Related to this point, the reviewer believes that the authors should refrain from using phrases like, “the mechanism is unknown”. The readers may find this claim disingenuous given the fact that the previous paper found many of the similar mechanisms (e.g. permeation of the mitochondrial membrane). The reviewer would suggest, instead, highlighting the contrast, or lack there of, between activity against HepG2 and A549 cells as previously described.
3. Some of the figures, including Figure 4, Figure 6, and Figure 7 are a little bit fuzzy, the reviewer would suggest making sure that the image is sharp. The copy editor will likely take care of this.
Technical:
1. Related to Style Point #1, the authors should cite that paper since it contains more convincing information about the lack of Hepatotoxicity and Nephrotoxicity (Figure 5 of that July paper). Some data or reference should be added to show that there are not significant off-target effects.
2. Could the authors please comment on the number of mice used in the study and if any of the mice died prior to extraction of the tumors? In other words, did every mouse survive the full two weeks? A survival curve would also be helpful to address Technical Point #1 as a supplement to Body Weight in Figure 9E.
Author Response
We greatly appreciate your careful review of our manuscript and all constructive suggestions. Point-to-point responses are attached as Word documents..

Reviewer 2 Report
In the manuscript ‘Smp24, a scorpion venom peptide exhibits a potent antitumor effects against Hepatoma HepG2 Cells via Multi Mechanisms In Vivo and In Vitro’ many experiments are reported to characterize the type of death induced by the scorpion venom peptide Smp24 in HepG2 cells, in vitro cultured, and grafted into the right flank of BALB/c nude mice.
Despite the large number of experiments, this work is not convincing because the results are contradictory and sketchy. Some experiments would show that the peptide induces a type of necrotic death (Figure 4), others that the peptide at the same concentrations and times induces apoptosis (Figure 6), but these two these two types of death cannot coexist in the same cells. Furthermore, some experiments would show that the peptide does damage to the cytoskeleton, others to the mitochondria, and others that it dysregulates the cell cycle, but in no case is there a demonstration of a direct effect of this peptide on these cellular structures or regulations.
Another important issue: the peptide is used at very high concentrations in in vitro experiments. Up to a concentration of 2.5 uM (u = micro), which corresponds to 6.5 mg/ml (MW c.a. 2.6 kDa IWSFLIKAATKLLPSLFGGKKDS), the peptide does not induce significant death even after 48 hours. The cell death is observed with concentrations of 5, 10 and even 20 uM (13, 26 and 54 mg/ml) and long times (12, 24 and 48 hours). At this concentration many peptides are toxic, the authors should use a scrambled control peptide with the same isoelectric point as Smp24, to see if the effect is specific. By comparison, melittin, another cationic antimicrobial peptide with toxic action on cancer cells, cited by the authors as an example of a peptide that is internalized in cells, induces HepG2 cell death at 24 h with an IC50 of 1.4 ug/ml (PMID: 34544028), a ten thousand times lower dose.
In experiments conducted on cells grafted into the flank of mice, however, a concentration of SMP24 of 2 mg/kg is used, which is much lower than that used on cells in vitro. Despite this lower concentration, respect to the in vitro experiments, the tumor masses created by HepG2 seem to be significantly reduced following treatment with the peptide. Why does this occur? Does the peptide have a different mechanism on grafted cells in mice?
There are many other problems to be noted in the manuscript, I list some of them below:
Fig. 1C e 1D: the time of action of the toxin is not reported
Figure 1Db shows that the cells after 24 hours of treatment detach to a considerable degree, while in Figure 2Bd the cells are still present and attached. Perhaps it depends by the amount of protein used (the quantity of toxin used in the internalization experiment is not reported), but since the toxin seems to enter the cells in such a massive way, how can the cells be still fine after 24 hours?
Figure 6C: the band of the representative western blot image do not correspond to the quantification reported in the histogram (see cleaved caspase 9, caspase 3 and PARP). The statistical significance (indicated by the stars number) of the data reported in this and in other histograms in many cases is improbable, as the same number of stars is reported over columns having similar and very different heights compared to the controls.
Author Response
We greatly appreciate your careful review of our manuscript and all constructive suggestions. Point-to-point responses are attached as Word documents.

Reviewer 3 Report
The authors examine the effects of the scorpion venom-derived peptide, SMP24, on cytotoxicity, cell membrane, cell cycle distribution, apoptosis and autophagy using HepG2 hepatoma cells.
There were minor grammar/syntax errors. Below are some examples, but this reviewer suggests to re-check the manuscript throughout. The authors should also refer to an excellent guide for writing: Strunk & White, Elements of Style.
Eliminate the use of colloquialism throughout the text, e.g., "What's (is) more", "Besides", etc.
Lines 43-44: "...while do not affect the normal liver cell."... "while not affecting normal liver cells."
Lines 51-53 (add bold corrections): "Smp24 also has cytotoxic effects against HepG2 hepatoma cells, as presented in ATP release assays; nevertheless, its mode of action remains unknown."
Line 118 (add bold corrections): "...control cells, which had flat microfilament control cells which had flat microfilament..."
Line 125 (change): in the presentce of Smp24.
Edit the run-on sentence at Lines 139-142 and 161-164.
Line 190 (change): "...the underlineing... apoptosis-inducinged..."
Lines 362-364 (change): Smp24 enters HepG2 cells via endocytosis and pore formation, causing mitochondrial dysfunctions and membrane defects; which consequently, resulting in cell necrosis, cycle arrest, apoptosis and autophagy."
Line 416 (change): "... after treatedment..."
Line 438 (bold): "In another set of experiments,..."
Author Response
We greatly thank you for your carefully reviewing our manuscript and mentioning these grammar/syntax errors. We have made corrections for all and some sentences were rewritten, too. The changes have been marked in different colour in our revision. Thanks you again!
Reviewer 4 Report
Reference: Manuscript submitted to TOXINS September 2022 “Smp24, a scorpion venom peptide exhibits a potent antitumor effects against Hepatoma HepG2 Cells via Multi Mechanisms In Vivo and In Vitro”
General comments: In the submitted manuscript, the authors report results of evaluation of the cytotoxic and antitumor activity on hepatoma cell line HepG2, of cationic antimicrobial peptide Smp24, found in the venom of Scorpio Maurus palmatus. According to the data, the anti-tumor activity of the peptide under analysis is associated with the induction of cell apoptosis, cycle arrest, and autophagy, by causing cell membrane disruption and mitochondrial dysfunction. After a careful reading of the manuscript, it is my opinion that it is a well-written text, the authors demonstrate knowledge of the subject discussed and the results are consistent with the conclusions. However, in my opinion some figures can be changed and some sentences can be rewritten in a revised version of manuscript. I forward to the authors some suggestions that may make the text more attractive and complete for readers in the area.
Specific Comments
1- In the line 9 and 47 … venom of Scorpio maurus palmatus. Please change maurus by Maurus with letter M in capital, and the names written in italics Maurus palmatus.
2- In the line 18 …. autophagy via membrane disruption and mitochondrial dysfunction,… I suggest to white …autophagy via cell membrane disruption and mitochondrial dysfunction, …. In order to avoid confusion with mitochondrial membrane disruption!
3- In the line 20 … Antimicrobial peptide. I suggest to change to antitumor peptide, based on it’s previously and currently described properties!
4- In the Introduction of text, among lines 36-38, the authors wrote… antimicrobial peptides (AMPs) can represent a potential therapy by selectively targeting tumor cells through binding the negative charged phosphatidylserine on their surface …. Although there is logic in this mechanism, since molecules with a positive charge, such as the peptides studied, can bind molecules with a negative charge, such as Phosphatidylserine, in the logic of membranes there is an enrichment of phosphatidylserine in the intracellular monolayer of cell membranes. How to explain the accessibility between these two molecules but in separate environments. One outside of the cell and one in the luminal monolayer? It would be interesting for the authors to explain this! Have the peptides size to enter in cells spontaneously? Or they have hydrophobic portions that allow membrane entering? Why not to interact with negatively charged Syndecans present in cell surface?
5- On the results shown in figure 1 that indicate cytotoxicity and inhibition of cell proliferation caused by the peptide Smp24. In Figure 1A, how do the authors explain that the inhibitory results of cell proliferation on HepG2, when 10, 15 and 20 micromolar of the peptide were used, were similar, not showing concentration dependence?
6- In Figure 1B, where the authors studied the cytotoxic activity of the Smp24 peptide on HepG2 cells, varying concentrations and exposure time, how can we explain that at different exposure times the results were practically identical, with no time-dependent toxicity? After 12 hours in the cellular environment, would the peptides be inactivated, since the times 12, 24 and 48 hours gave practically the same signs of cytotoxicity for different concentrations?
7- In figure 1D, where the authors studied morphological changes of HepG2 cells. The images were taken at a magnification of 100X, and this is good to have a panoramic idea of ​​the effects of the treatment, but to have an idea of ​​morphological changes it would be good to show the images in magnifications larger than 400X or even 1000X, where details of the cells could be seen. Authors could include inserts of signals of morphological changes. Also why did the authors not show the normal hepatocyte cell lineage LO2, as done in Figure 1A?
8- Also about the text written on the cytotoxicity results in figure 1, lines 60-61 and lines 276 and 277…While compared with the slight effect on LO2 cells, Smp24 presented a remarkable inhibitory effect on the proliferation of HepG2 (Figure 1A) or at discussion chapter. I suggest that the authors substitute the words …slight effect on LO2 cells and replace it with ...lower or minor effects on LO2 cells compared to HepG2 cells ...because in my opinion the citotoxic effect on LO2 cells was concentration dependent and significant as changed viability from 100 to 40% of cells under the conditions used.
9- About figure 2. If the positively charged Smp24 peptide enters the cell and has the potential to interact with negative molecules and structures as phosphatidylserine in the cell membrane, why does it not interact with the nucleus, which is the cellular center rich in negative groups, as a function of the high concentration of phosphate in the nucleotides of DNAs and RNAs as do for instance Histones also rich in positive amino acids ?
10- Between lines 85 and 88 the authors wrote … the zeta potential of HepG2 cells treated with Smp24 (2.5, 5, and 10 µM) was increased from -14.67mV to -8.11 mV in a concentration-dependent manner, indicating that the changes in cell surface charge were caused by the binding reaction between Smp24 and HepG2 cell surface. The changes in Zeta potential under the conditions analyzed are clear and really significant, but the experiment did not allow the authors to conclude that these changes are due to peptide bonds on the surface of the cells tested, because this was not shown! These alterations may be due to other intracellular alterations after peptide entering that culminate in alterations of the Zeta Potential, but to show that there was a binding of the studied peptide on the cell surface, the authors would have to perform a flow cytometry reaction marking the toxin in cells not permeabilized, or confocal microscopy with antibodies recognizing the peptide and showing colocalization on the cell membrane, or transmission electron microscopy and labeling with colloidal gold for colocalization of peptide and cytoplasmic membrane for instance.
11- Could these changes in Zeta Potential also be consequences of the disruptive activity of the peptide in the plasma membrane? Since opening holes for example in the cell membrane this could change solutes/water in and out, and the entire electrical potential on the cell surface!
12- About figure 2B, I would like to see a reaction for the labeling of the nucleus, but where the authors did not label the cells with DAPI, to verify if there is labeling of the cell nucleus, which as already emphasized, is an organelle rich in molecules with negative charges (DNAs and RNAs).
13- As the authors have the target peptide of the studies (Smp24) labeled with fluoriscein, why not do a confocal microscopy, to finally prove if the peptide binds to the plasma membrane of cells?
14- As the authors postulate that there is an alteration in the integrity of mitochondria after treatment of cells with the peptide Smp24, it would be interesting to perform a double-labeled confocal microscopy with some fluorescent marker for mitochondria as Mitotracker for instance. This would show peptide and organelle colocalization!
15- Figure 2C brings us some uncertainties! For example, how can the authors state that the Smp24 peptide binds to phosphatidylserine molecules in the intraluminal monolayer of the plasma membrane, if in the extracellular monolayer there are several polyanionics such as Sindecans (heparan-sulfate proteoglycans), which could already interact with the positive peptide Smp24 binding this on the surface of cells.
16- Between lines 101 and 102 the authors wrote …. These findings suggested the internalization of Smp24 into HepG2 cells via pore formation and endocytosis. As also in conclusion …. Smp24 enters HepG2 cells via endocytosis and pore formation line 362. This statement appears to be a paradox in cell biology, as pore formation resembles a hole in the membrane, with actively entry into the cells. Endocytosis, on the other hand, resembles clathrin-coated pits and a cell surface receptor that has a transmembrane sequence to bind adaptin and crathrin e formation of pits. They seem to me to be mutually excluding mechanisms. I would like the authors to talk about this.
17- The authors never thought about the existence and description of a receptor for this peptide on the surface of cells. A Sidecan for example. This could be studied, for example, with a treatment with heparitinase in the cells, which should inhibit or hinder the entry of the peptide into the cells.
18- Also if the peptide enters via receptor-dependent endocytosis, it will initially stay in endosomes and then go to lysosomes, intracellular pathway of endocytosed materials and finally be digested in the intra-lysosomal environment. How then the interaction with mitochondria?
19- The interesting thing about the experiment carried out by the authors to study the action of the Smp24 peptide on the actin cytoskeleton is that the cells were not rounded, but lost the fluorescence signal in red. If there was depolymerization of actin filaments into actin monomers, the cell would change its shape from a spread shape on the substrate as in the control, to a round shape. What the result shows was a decrease in fluorescence. Also, It appears that the number of DAPI-labeled nuclei decreased dramatically in the comparison with control and 10 micromolar peptide, suggesting that the cells had detached from the substrate. This cell substrate detachment could indicate the inhibition of receptors present on the cell surface, and a good example is the negatively charged Sidecans, ligands of the Extracellular Matrix components. I Would like the authors opinion about this?
20- In Figure 4A, in the treatment of cells with the peptide Smp24 for 12 hours, apparently there was no concentration-dependent activity for 5, 10 and 20 micromolar, however, for times 24 and 48 there was! Could the authors explain this discrepancy?
21- Still in figure 4B, the authors show morphological alterations of cells treated by the Smp24 peptide. Cells are shown by scanning electron microscopy. However, the authors only show a single cell per figure, when they should show several cells in the same field (a panoramic view) to really prove the generalized action of the peptide. Thus, it gives the impression that the cells were chosen and this may give rise to doubts about the disruptive action of the peptide. In a revised version, this figure could be replaced by a more panoramic image, proving the disruptive action of the peptide and an insert can be added according options of authors.
22- About figure 5, I would like more details on materials and methods, as done for the other methodologies. In this case, the authors do not present methodological details of how the experiments to prove the direct action of the Smp24 peptide on the mitochondria were carried out.
23- I would feel more confident if the authors added to the revised version a panoramic view of a transmission electron micrograph, showing mitochondrial damage as argued.
24- Throughout the text between lines 158 to 169, it would be good for the text, if authors to add more details in the revised version about the mechanisms and objectives of the treatments and dyes used to verify changes in mitochondrial functions.
25- About figure 6, I would like to see more details in the legend, especially figure 6B.
26- Regarding the statements written between lines 182 to 184, I believe that figure 6A, although it shows a change in the labeling profile with DAPI, does not show in detail what was written in the text. If the authors really want to show the morphological changes as wrote, It would be interesting to show a transmission electron micrograph, where the ultrastructural details of nuclear alterations could be seen, as stated.
27- Some authors question the use of Annexin-V to mark apoptosis in situations where the cell membrane is altered, since the mechanism of Annexin V to recognize phosphatidylserine in the apoptotic extracellular monolayer of the plasma membrane can be misinterpreted by the entry of Annexin V through the holes originated by the treatment, and thus recognize phosphatidylserine within the cell. I would like the author’s opinion on this fact.
28- In the line 194 the authors wrote … cleared-caspase 3, or they want to say cleaved-caspase 3?
29- Still on figure 6, although it seems clear to me that there were changes in several markers correlated with cell apoptosis, for figures 6C and 6D, some densitometries seem to be discrepant in relation to WB! For example for cleaved caspase 3 and cleaved PARP. I would like the opinion of authors?
30- About figure 7, I would like further clarification from the authors throughout the results and discussion of the text. Figure 1 shows proliferation inhibitory activity of HepG2 cells treated by the Smp24 peptide. On the other hand, the data shown in Figure 7 indicate that after treatment with the Smp24 peptide, in a dose-dependent manner, greater cell population in S phase, which indicates an increase in DNA synthesis, in addition to more cells in G2/M, which indicates cell division, and thus greater proliferation. Figure 1 and Figure 7 seem to me to be discrepant. I Would like to see the authors' opinion?
31- In the line 228 the authors wrote … s a potential cancer therapy for autophagy… In my opinion better to change …a potencial cancer cellular control.
32- On figure 8, once again, it would be interesting for the authors to show a transmission electron micrograph where cellular alterations characteristic of autophagy, such as autophagosomes and autophagolysosomes, could be seen in the cytoplasma of cells.
33- In addition, there are fluorescent dyes that indicate autophagosomes and autophagolysosomes in the intracellular environment. This labeling would be very interesting and would confirm the hypothesis raised by the authors!
34- About figure 9, the authors could detail more in the legend, the experimental schedule fig.9A. The data on the decrease in tumor mass and size are very interesting, and undoubtedly confirm the antitumor potential of the peptide under analysis. However, histopathological analyzes fig. 9G need to be changed, because in the amplifications shown it is not possible to see the statements written throughout the text, lines 253 to 255. In the modified version, add inserts and larger amplifications of the figures and arrows or arrow-heads to point such as histopathological changes!
35- Still related to the antitumor activity experiments, it would be good for the authors to show biochemical markers of renal, hepatic and hematological functions, to finally prove that the treatment with peptides does not have, or has low toxicity for animals.
36- At lines 281 to 283 the authors wrote ….whereas cancer cell membrane consists of exclusive anionic constituents like phosphatidylserine, O-glycosylated mucins, sialylated gangliosides and heparan sulfate. I would like the authors to cite references to this sentence, because it undoubtedly brings controversy, especially regarding the presence of phosphatidylserine, on cell surface, which is a marker of cell apoptosis!
37- The authors talk about the biotechnological potential of the use of this peptide as a pharmacological agent used directly in the treatment of liver cancer or other tumors. But even though it is a low molecular mass peptide, it can generate an immune response and mainly neutralizing humoral (antibodies), inactivating the peptide after sequential uses and mainly after third or more doses. Also, as a peptide, body proteases can degrade the molecule avoiding a pharmacological action. How to overcome the humoral response performed by patients after successive doses of the peptide and proteolytic activity of body proteases?
38- Undoubtedly, tumors in the liver, pancreas and stomach have a poor prognosis and are considered big killers. Any successful therapy novelty in this field deserves respect and investment. Nowadays, in order to overcome tumor resistance to chemotherapy, many researchers have tried to use several drugs, with different mechanisms and at the same time, for the treatment of big killer tumors. Did the authors ever think about using this peptide as an adjuvant or as a sum of already standardized drugs in anti-tumor chemotherapy?
Author Response

(The authors gave the same response as above.)

Round 2
Reviewer 2 Report
The answers of the authors are sufficient
Reviewer 4 Report
After careful reading of the corrected version of the manuscript and the response letter written by the authors, it is my opinion that this revised version can be published by TOXINS. In the response letter, the authors show technical competence and knowledge of the topics raised by the reviewer. Although some suggestions indicated by the reviewer were not incorporated, as the authors argue for lack of equipment and resources, many other suggestions were incorporated and made this revised version more robust, attractive and technically complete. The text of the revised manuscript has also undergone several changes making this revised version more attractive and complete. As commented, new discoveries that will bring advances in the inhibition of tumors considered big killers must be treated with empathy, and I sincerely hope that the authors succeed in their next experiments, many of which are suggested in my comments, to advance in the discoveries with this toxin. Best regards.